# Morphological Reconstruction of a Critical-Sized Bone Defect in the Maxillofacial Region Using Modified Chitosan in Rats with Sub-Compensated Type I Diabetes Mellitus

**DOI:** 10.3390/polym15214337

**Published:** 2023-11-06

**Authors:** Nadezhda N. Patlataya, Igor N. Bolshakov, Vladimir A. Khorzhevskii, Anatoli A. Levenets, Nadezhda N. Medvedeva, Mariya A. Cherkashina, Matvey M. Nikolaenko, Ekaterina I. Ryaboshapko, Anna E. Dmitrienko

**Affiliations:** 1Department of Fundamental Medical Disciplines, Institute of Medicine and Biology, Faculty of Medicine, State Educational Institution of Higher Education, Moscow State Regional University, Moscow 105005, Russia; nadya_barahtenko@mail.ru; 2Department Operative Surgery and Topographic Anatomy, Voino-Yasenetsky Krasnoyarsk State Medical University, Krasnoyarsk 660022, Russia; 3Department Pathological Anatomy, Voino-Yasenetsky Krasnoyarsk State Medical University, Pathological and Anatomical Department Krasnoyarsk Clinical Regional Hospital, Krasnoyarsk 660022, Russia; vladpatholog@yandex.ru; 4Department Surgical Dentistry and Maxillofacial Surgery, Voino-Yasenetsky Krasnoyarsk State Medical University, Krasnoyarsk 660022, Russia; aalevenets@mail.ru; 5Department of Human Anatomy, Voino-Yasenetsky Krasnoyarsk State Medical University, Krasnoyarsk 660022, Russia; medvenad@mail.ru; 6Pediatric Faculty, Voino-Yasenetsky Krasnoyarsk State Medical University, Krasnoyarsk 660022, Russia; mashacherkasha@mail.ru (M.A.C.); e.ryaboshapko@yandex.ru (E.I.R.); anna.e.dmitrienko@gmail.com (A.E.D.); 7Department of Maxillofacial and Plastic Surgery, Moscow State University of Medicine and Dentistry, Moscow 127473, Russia; kopm980@mail.ru

**Keywords:** modified chitosan, polyelectrolyte complex CH–SA–HA rats, type I diabetes mellitus, critical-sized bone cavity, morphological reconstruction, histomorphometric criteria

## Abstract

It is known that complexes based on natural polysaccharides are able to eliminate bone defects. Prolonged hyperglycemia leads to low bone regeneration and a chronic inflammatory response. The purpose of this study was to increase the efficiency of early bone formation in a cavity of critical size in diabetes mellitus in the experiment. The polyelectrolyte complex contains high-molecular ascorbate of chitosan, chondroitin sulfate, sodium hyaluronate, heparin, adgelon serum growth factor, sodium alginate and amorphous nanohydroxyapatite (CH–SA–HA). Studies were conducted on five groups of white female Wistar rats: group 1—regeneration of a bone defect in healthy animals under a blood clot; group 2—regeneration of a bone defect under a blood clot in animals with diabetes mellitus; group 3—bone regeneration in animals with diabetes mellitus after filling the bone cavity with a collagen sponge; group 4—filling of a bone defect with a CH–SA–HA construct in healthy animals; group 5—filling of a bone defect with a CH–SA–HA construct in animals with diabetes mellitus. Implantation of the CH–SA–HA construct into bone cavities in type I diabetic rats can accelerate the rate of bone tissue repair. The inclusion of modifying polysaccharides and apatite agents in the construction may be a prospect for further improvement of the properties of implants.

## 1. Introduction

Polycationic polysaccharide polymer chitosan creates a solid frame of wound coating, entering into physical synthesis with natural anionic polymers and providing antibacterial and antitoxic effects in bone cavity. It is a building material and regulator of the formation of highly organized regeneration of connective tissue and a non-toxic, highly biocompatible material, completely biodegrading in the wound cavity with the transition to a gel mass. Sodium alginate is a polyanionic polysaccharide polymer, forming a strong electrostatic (physical) structure with chitosan. It is biodegradable, non-toxic and highly compatible with biological tissues. It is a building material for bone wound regeneration and creates chemical equilibrium when included in the composition of medicinal products, increasing the elasticity of the polymer structure.

In the presence of a modified chitosan polyelectrolyte complex, early signs of angiogenesis and the formation of a new bone are recorded, starting from 7 days after the experimental formation of a bone defect [1]. Filling of an extensive bone cavity with chitosan ascorbate-alginate sodium-hydroxyapatite (CH–SA–HA) containing quaternized chitosan (CH), sodium salt of alginic acid (SA) and nano-structured hydroxyapatite (HA) showed that by the end of the 4th week, 80% closure of the defect was recorded on the tomogram, and after 8 weeks, it reached 100%. Ten weeks after the injury, there are no signs of a bone defect. The analysis registers dense bone corresponding to the density of healthy bone. The bone density in the experimental group was 810–1050 HU for the spongy substance, and for the compact part, it was 1300–1500 HU, which is higher than in the control group.

Lack of insulin, hyperglycemia and the accumulation of glycation end products (AGE) support systemic chronic inflammation with microvascular damage. The combination of these factors violates the architecture and biomechanical properties of bone [2,3], changing the ratio of the main mineral component of hydroxyapatite and the organic component—collagen type I [4]. Overproduction of peroxides and deactivation of catalase leads to differentiation and proliferation of osteoclasts [5,6,7], increasing the concentration of H_2_O_2_ in osteoclasts and reducing the expression of antioxidant enzymes, This process repeatedly accelerates the differentiation and survival of osteoclasts as well as their proliferation [8]. On the contrary, bone resorption is accompanied by apoptosis of osteoblasts [9,10,11], i.e., a decrease in their differentiation and activity [12,13]. Such pathophysiological shifts in periodontal tissues weaken the girder structure of mineralized bone [2,14,15,16]. This process leads to a deterioration in the mechanical properties of the bone [17] and significantly complicates bone regeneration, especially in the presence of a bone cavity of critical size. Morphological analysis indicates the destruction of the alveolar bone in the periapical and furcal regions and an increase in the areas of accumulation of osteoclasts with signs of bone lysis [18].

A characteristic morphological sign of reduced bone regeneration is a decrease in the thickness of the bone trabecula [19,20]. The rate of reduction of the bone callus in which cartilage is formed, the rate of cartilage resorption in the bone defect [21], the size of the newly formed bone [22] and the rate of formation of the vascular network [23] should be considered as important morphological signs of the state of bone regeneration in the analysis of the defect, requiring calculation in morphometric analysis. Natural polysaccharides such as chitosan and sodium alginate are today considered promising materials for closing a bone cavity of critical size. These materials work best when used for bone regeneration [24,25,26,27,28]. The similarity of the molecular structure of chitosan, sodium alginate, chondroitin and hyaluronic acids and heparin sulfate and the mutual improvement of their own biological properties in relation to tissue regeneration dictates the need to create copolymers with high regenerative capacity [29]. The high molecular weight of the polymer and the high positive charge reduce the degradation rate of the matrix base, creating a sufficient time period in which to eliminate the bone cavity of critical size. The molecular and functional similarity of chitosan with highly hydrated glycosaminoglycans (GAG) of connective tissue and the ability of anionic chemical groups (SO_3_, COOH-, OH-) to quaternize the chitosan molecule and collapse it to nanoscale through the tissue compartment, which gives grounds for using polyelectrolyte chitosan complex for delivery of building material and elimination of a bone cavity of critical size [30]. This conformation of chitosan was obtained in [31]. It was noted that the globular shape of a polymer molecule with a molecular weight of 1000 kDa is in the range of 48–52 nm. Under weak acidic conditions, the rigid rod of the molecule transforms into a helix. The inclusion of sulfated anionic polymers into the additionally protonated amno group (NH_3_) of the chitosan molecule twists the linear polymer into a globule. This process happens quite quickly. The more sulfo groups a polymer molecule contains, the more efficient the conformation process is (for example, a chondroitin sulfate molecule has one SO_3_ group, and a heparin sulfate molecule has three SO_3_ groups). At the same time, the pH of the medium increases from 1.55 to 3.5 and reduces the size of nanoparticles by 4.2 times. This process is very important in the bone cavity of the maxillofacial region, since the rapid penetration of the structure into the defect area quickly starts the process of vascular endothelialization of the matrix [32]. The very small size of the polyelectrolyte complex enhances platelet adhesion, which stimulates the functions of osteogenic cells [33] and increases good contact of the polymer gel with the newly formed and maternal bone [34]. This reconstruction of chitosan can activate the proliferation of vascular endothelium and the endothelization of the walls of the bone cavity and the polysaccharide structure itself, triggering the proliferation and differentiation of osteoblasts. The task when using a biodegradable three-dimensional frame is to maintain the function and control the degradation of the polymer in the bone tissue until the full formation of a new bone. To stimulate bone neoplasm in diabetes mellitus and increase tissue strength, it is recommended to include hydroxyapatites in the complex [35,36,37]. Nano-sized hydroxyapatite, which ensures the strength of bone tissue, is located between bundles of collagen fibers in the form of parallel-oriented plates and increases the synthesis of its own bone tissue. The inclusion of hydroxyapatite in the chitosan matrix is designed not only to enhance the strength characteristics of the implant, but also for the early formation of native bone [38].

The inclusion of angiogenic and osteogenic growth factors in the chitosan matrix accelerates the formation of a full-fledged bone. For example, preliminary studies have shown that the addition of the low-molecular-weight serum growth factor “adhelon” to the collagen-chitosan structure increases the inclusion of ^3^H-thymidine in the primary culture of proliferating mouse fibroblasts cultured on various substrates for 16 h [39].

The end result of molecular transformations is a decrease in the induction of osteoclastogenesis molecules, with a decrease in bone lysis zones and an increase in the volume of newly formed bone surrounding polysaccharide implants.

The result of molecular transformations is the active differentiation of osteoblast precursors, the formation of a large mass of osteocytes filling a bone defect of critical size and microvascular endotheliocytes in the Havers channels [1]. For the hydration of chitosan molecules and the survival of osteoblasts, ascorbic acid is also included [39].

The concept of this study is based on the ability to form an early osteogenic reaction in the maxillofacial region using modified polysaccharides, despite the active development of diabetes mellitus and the presence of a critical-sized bone defect.

Thus, to confirm the formation of conditions for the intensive formation of young bone in diabetes mellitus—early covering of bone beams with a mass of osteoblasts and the formation of Havers channels with a high content of endotheliocytes—the authors of the work used a polyelectrolyte complex based on quaternized high-molecular chitosan with an additional degree of amination and a degree of deacetylation of 95%. The interest is a detailed morphological analysis of the bone cavity under the conditions of filling using a polysaccharide design.

## 2. Materials and Methods

The work was performed following ethical principles established by the European Convention for the Protection of Vertebrate Animals used for Experimental and Other Scientific Purposes (Strasbourg, 18 March 1986, adopted on 15 June 2006). All manipulations with the animals were performed following the regulations specified in the Guide for the Care and Use of Laboratory Animals (National Research Council, 2011). The work was approved with the complex scientific theme No. 01201362513 (1 January 2013–1 January 2021), “Fundamental and applied scientific and technical developments of nano-level biopolymer structures and technologies for their production for use in cell and tissue engineering for socially important human diseases”; section “Dentistry”: Obtaining, testing and introduction into clinical practice of cell substrates for direct implantation into hard and soft tissues of the periodontium in order to reconstruct the tissues of the maxillofacial region and eliminate the causes of the formation of degeneration zones and periodontolysis zones; research topic: “Restoration of the structure of the bone tissue of the maxillofacial region using polysaccharide polymers with extensive traumatic defects in conditions of sub-compensated diabetes mellitus.” Bioethical Commission for Working with Animals by the Ethics Committee of the Voino-Yasenetsky Krasnoyarsk State Medical University of the Ministry of Health Russian Federation (Protocol No 2 of 28 October 2019).

### 2.1. Composition and Production of Modified Chitosan

“CH–SA–HA” gel mass containing a 2% solution of chitosan ascorbate (CS) (dissolution of the polymer in ascorbic solution acid in a ratio of 1:1.5) with a molecular weight of 695 kDa and a degree of deacetylation of 95% (a special purified chitosan obtained in Vostok-Bor-1, Dal’negorsk, Russia); specifications (No. 9289-067-004721224-97) including, per 1 g of dry chitosan ascorbate, 100 mg of sodium chondroitin sulfate (Sigma, Kawasaki, Japan), 100 mg of sodium hyaluronate (Sigma), 2,5-5 mg of heparin sulfate (Russia, Pharm.Art., Redfern, Australia, (No. 42-1327-99)), 110 mcg/g serum growth factors in cattle “adgelon” (SLL “Endo-Pharm-A”), Moscow region, Schcholkovo, Russia, specifications (No. 113910-001-01897475-97), 4% sodium alginate (SA) (Pharm.Art. No. 42-3383-97) or specifications (No. 15-544-83; Arkhangelsk Algal Plant. Co., Moscow, Russia), including 50% amorphous hydroxyapatite (HA) (5–20 nm, Russia, Pharm.Art. (No. 42-3790-99 or GOST 12.1.007-76)), with a ratio of chitosan ascorbate to sodium alginate of 1:1 [40].

An amount of 7.2 g of chitosan was added to the prepared solution of ascorbic acid (Pharm. Art. No. 42-2668-95) with stirring at a temperature of +20–22 °C; the mass was stirred for 4–5 h until the chitosan was completely dissolved. Aqueous solutions of sodium salts of chondroitin sulfuric acid, hyaluronic acid and heparin were successively added to the resulting 4% chitosan solution with constant slow stirring using a magnetic stirrer in a total volume equal to the volume of chitosan ascorbate. The introduction of each subsequent ingredient was carried out after the homogeneous mixing of the previous one with the chitosan gel. As a result, a 2% chitosan poly-ionic complex was obtained. Next, a 4% aqueous solution (gel) of sodium alginate was prepared, and 50% (by dry weight) of hydroxyapatite was added. The finished chitosan solution was thoroughly mixed with the sodium alginate solution using a high-speed mixer [41].

The gel mass was poured into 2 mL vials and placed in a LZ-45 Frigera (Kolín, Czech Republic) sublimation unit at a plate temperature of +20–25 °C. The mixture is frozen to a temperature of −40 ± 3 °C. Freezing time is 3–4 h, then the chamber is put under vacuum to a residual pressure of 133 Pa (1 mm Hg), after which the process of drying the product begins due to the supply of heat from the coolant to the frozen product through the heating plate surface. The sublimation process is carried out at a residual pressure of 44.3 to 133 Pa (0.33 to 1 mm Hg) and a coolant temperature of 45–50 °C. The temperature of the product should not exceed +50 °C. The complete drying cycle of the product is 8 ± 2 h. Vials with lyophilized mass are hermetically sealed, and gamma sterilization is carried out at a radiation dose of 5–7 kGy.

### 2.2. Animal Characterization and Modeling of Type I Diabetes Mellitus in Rats

For the study, white female Wistar rats (supplier LLC BioTech, Moscow, Russia, weight 200–250 g, age 3 months at the beginning of the experiment) were used. The studies included 5 groups of animals, 10 rats in each group: group 1 (control No. 1, healthy animals, comparative analysis, regeneration of the bone cavity under a blood clot); group 2 (control No. 2, animals with a model of sub-compensated diabetes mellitus, regeneration of the bone cavity under a blood clot); group 3 (control No. 3, animals with a model of sub-compensated diabetes mellitus, regeneration of the bone cavity when filling a bone defect with a lyophilized collagen sponge, type I polymer, product purity 99%; “Bioactive collagen”, SLL “LAONA LAB” certificate of conformity No. POCC RU.PK 08. H00305; technical conditions 10.11.60-001-64516066-2017; test reports No. D0317m dated 28 November 2017; No. D0516 dated 28 November 2017 (Joint-Stock Scientific Society “Perfumetest” testing center)); group 4 (control No. 4, healthy animals, regeneration of the bone cavity when filling with lyophilized CH–SA–HA mass); group 5 (experienced group, animals with a model of sub-compensated diabetes mellitus, regeneration of the bone cavity during filling of a bone defect with a lyophilized mass of CH–SA–HA) [1]. In each group, during morphological analysis, peripheral zones remote from the bone cavity were identified, and a comparative analysis of bone regeneration in the central and peripheral zones was performed.

Destruction of pancreatic beta cells was achieved with a single subcutaneous injection of pure alloxan hydrate (C_4_H_2_O_4_N_2_·3H_2_O; Mm 196.17, Chemapol, Praha, Prague, Czech Republic) at a dose of 35 mg/rat against a preliminary 48 h fast.

Peripheral blood sugar levels were determined using an EasyTouch GCHb (Bioptik Technology, Inc., No. 188 Jhonghua South Road, Jhunan Township, Miaoli County, 35057 Taiwan) glucometer. The period of observation of animals after the development of type I diabetes mellitus was 30 days. After the steady development of a state of hyperglycemia for 10 days at a peripheral blood sugar level above 10 mmol/L, after modeling, daily subcutaneous insulin administration at a dose of 2 ME was added (Biosulin P soluble human genetically engineered; Open Joint Stock Company Pharmstandard-UfaVITA, Ufa, Russia).

### 2.3. The Conditions of Animal Detention

The conditions of biological test systems in the CDI CI correspond to the Guide for the Care and Use of Laboratory Animals, 8th edition. Washington (DC): National Academies Press (US); 2011. USA. Animals were managed in individually ventilated cells made from polysulfone Sealsafe, 461 × 274 × 228 mm (production TECHNIPLAST.P. A.). The rooms, which contain biological test systems, had a controlled temperature of 18–24 °C, humidity of 30–70%, illumination for 12/12 h and a multiplicity of air (XII without recirculation). Control of climatic parameters was carried out in accordance with the SOP “control of climatic parameters in the premises of the vivarium.” Distribution of feed and water was carried out at a fixed time: litter was changed once a week in accordance with the SOP “preparation of cells for biological test systems. Marking. Change of bedding, feed, water”.

### 2.4. Modeling Defects of Critical Size in Rats

Under general intramuscular anesthesia with a mixture of Zoletil 100 at a dose of 3 mg/rat and Rometar at a dose of 0.5 mg/rat and under aseptic conditions, the skin in the lower jaw was treated with an alcohol solution of chlorhexidine. An incision was made in the inferior alveolar process region, 1.5–2 cm long, and the masseter muscle was bluntly separated along with the periosteum; the bone was exposed in the angle area. On the alveolar process, using a drill and a spherical bur, a rounded three-walled defect 4 × 5 × 4 mm in size was created. During surgical intervention, the bone was “cooled” with an aqueous solution of chlorhexidine using boron. The cavity of the bone defect was drained with tampons and filled with a CH–SA–HA lyophilic mass, a collagen sponge or an auto-blood clot. The defect was closed with periosteum, and the skin was sutured with separate 5.0 monofilament sutures. The wound was treated with an alcohol chlorhexidine solution.

### 2.5. Postoperative Period

Within 3 days after the intervention, the animals received Tramadol anesthetic solution 0.2 mg 2 times daily. During the first 24 h, animals were given water. Feeding was performed 24 h after the intervention solely with a mixture of “Polyproten-nephro” (SLL “Protenpharma”, Golikovo village, house 120, Solnechnogorsk district, Moscow region, 107 564, Russia) based on Supro XT 219 DIP Isolated Soy Protein with less than 2% Lecithin D 14084043 (“Solae LLC” Routes 47 and 9, Gibson City, IL, 60936, USA) for 3 days. Drug support was provided with a broad-spectrum antibiotic (ceftriaxone at a dose of 8 mg/rat). Animals were kept in a postoperative room with ultraviolet air sterilization and forced fine-cleaning system circulation. The skin sutures were removed on the 7th day after the intervention.

### 2.6. Morphological Analysis of Bone Tissue

The lower jaws were subjected to decalcification in a solution based on the disodium salt of ethylene-diamine-tetra-acetic acid (EDTA) until completely softened. The composition of the decalcifying solution was as follows: 250 g of EDTA was used per 1000 mL of distilled water with the addition of 50 mL of 40% sodium hydroxide solution. For the manufacture of histological preparations, blocks were cut out from the lower jaws of rats containing the area of a postoperative bone defect. The preparations after dehydration in ascending concentrations of alcohols were embedded in paraffin blocks. Using a MicroTec CUT4050 microtome, serial sections (20–25 sections) were made in the transverse plane (relative to the animal’s body) through the entire area of the bone defect. In addition, 2–3 cuts were made along the periphery of the bone defect. The prepared sections with a thickness of 4–5 µm were stained with hematoxylin-eosin (H-E). The sectioning procedure is shown schematically. The study design is shown in Figure 1.

The morphological study was carried out with scanning histological preparations in a FLASH 250 3D HISTECH histoscanner (Budapest, Hungary). Plain microscopy was performed in direct and polarized light of the area of the postoperative bone defect and the peripheral zone with an assessment of the histoarchitectonics of the bone tissue. A semi-quantitative assessment was performed to determine the nature of the distribution of foreign material in the bone cavity. The cellular reaction was studied, as was the state of the soft tissue component. Histomorphometric evaluation was performed on digital micrographs, which were obtained using CaseViewer Ver.2.6 3D HISTECH software.

For an objective histological assessment of bone tissue repair, criteria with a standardized nomenclature were used [42]:(BV)—Volumetric density of bone tissue, the percentage ratio of the volume occupied by bone structures to the total volume of the histological section;(BTT)—Thickness of bone trabeculae (mm). The criterion stipulates that the bone trabecula is a thin plate. Measurements were taken between the edges of the bone trabecula (5–8 measurements in relation to each trabecula with the calculation of the median);(ITS)—Intertrabecular spaces (mm), the distance between the edges of the cancellous bone trabeculae. The calculation is made in accordance with the so-called parallel plate model: BV minus BTT;(OBS)—Osteoblastic surface of bone trabeculae, the percentage ratio of the surface of bone trabeculae occupied by osteoblasts to the total bone surface;(OS)—Osteoid surface of bone trabeculae, the percentage ratio of the surface of bone trabeculae occupied by osteoid to the total bone surface, which was assessed by polarized light microscopy;(ES)—Eroded (osteoclastic) surface of bone trabeculae, the percentage ratio of the surface of bone trabeculae with the formation of gaps to the total bone surface, including the surface occupied by osteoclasts;(TBS)—Total bone surface;(FS)—Free surface of bone trabeculae, the percentage of the non-eroded surface of bone trabeculae and the surface not occupied by osteoblasts, osteoclasts to the total bone surface);

The assessment of histomorphometric parameters was performed in the area of the applied postoperative lower jaw bone defect, as well as in the area connected to the bone defect (3 mm from the edge of the bone defect). When analyzing the histomorphometric parameters of bone tissue in the periphery region of the bone defect, the authors did not divide the measurements into groups depending on the type of implanted material. Pilot studies did not show the significance of the influence of the implanted material type in the area of the bone defect on the average (median) histomorphometric values and the nature of the distribution of variable values in the peripheral area. Thus, the histomorphometric values of the peripheral area constituted an additional control group. The number of histological sections of each study group corresponded to the sensitivity (1 − β) − *p* = 0.75–0.8 (75–80%) for the estimated mean and standard deviation. The presented threshold value was chosen by taking into account the relatively small sizes of the studied samples. In this context, it should be noted that it is admissible to study small sample sizes, given that the relatively low sensitivity of such samples can be compensated for by more pronounced differences [43]. Thus, the presented acceptable sensitivity (1 − β) in the used samples was compensated for by a relatively low threshold value of the error of the first kind (α) − *p* = 0.01.

### 2.7. Statistical Analysis

Statistical analysis of data and the creation of graphic illustrations were carried out using the free software computing environment “R, version 4.2.1” and the programming language “R”. The assessment of the obtained variables in relation to compliance with the normal (Gaussian) distribution was carried out on the basis of the Shapiro–Wilk test, as well as on the basis of the graphical method (quantile–quantile plot). Most of the variables obtained obeyed the normal distribution law, and the cases of deviation of the variables from the normal distribution were not pronounced.

For variables deviating from the normal distribution, the following methods of data transformation were used to achieve compliance with the normal distribution: with right-sided (positive) skewness, square roots were taken from the obtained values, and in the case of left-sided (negative) skewness, the following formula was used: $\sqrt{max\left(x+1\right)-x}$, where “x” is the value obtained. Descriptive statistics of the obtained data were presented as median, 25% and 75% quartiles (Me[Q1;Q3]). During the choosing of statistical tests for assessing the type I error (α) and the sensitivity of the criterion (1 − β), parametric methods of statistical analysis were used: one-way analysis of variance (ANOVA); for paired comparisons of independent variables, Welch’s *t*-test was used; for multiple comparisons, we used Bonferroni amendment. The presented variants of assessments were carried out taking into account the equality of variances, the variables under study, as well as the level of sensitivity of the criteria not lower than *p* = 0.75. To assess the error of the first kind, taking into account the small volume of the studied samples, the threshold value *p* = 0.01 was used.

## 3. Results

### 3.1. Morphological Analysis of the Bone Cavity Walls in Induced Type I Diabetes Mellitus Development

The state of hyperglycemia in rats for 30–40 days leads to noticeable changes in the mechanisms of bone formation under conditions of bone cavity regeneration of a critical-sized defect. Multiple sections of bone tissue make it possible to obtain reliably distinguishable results on almost all morphometric criteria, both directly in the area of surgical intervention and on the periphery of the lower jaw.

A comparative analysis of the BV parameter of the interest zone between healthy animals (control 1), taken out of the experiment on the 4th week, and animals with sub-compensated diabetes mellitus (control 2) showed a significant predominance of the value of this indicator in healthy animals in the jaw defect area (in all possible options for intergroup comparison) and in the peripheral zone (*p* < 0.01). In control 1, a reaction of active bone lysis is characteristic, as is a high number of osteoclasts on the bone beams (Figure 2A). Comparison of the thickness of the bone trabeculae of the interest region in healthy animals and animals with sub-compensated diabetes mellitus showed significantly higher BTT values in healthy animals both in the area of the jaw defect and in the peripheral zone (*p* < 0.01). Comparison of the intertrabecular spaces’ sizes in controls 1 and 2 showed significantly lower ITS indices in healthy animals both in the area of the bone defect (for all possible options for intergroup comparisons) and in the peripheral zone (*p* < 0.01). OBS in the area of the jaw defect of healthy rats (control 1) after 4 weeks of the experiment significantly exceeded the same indicator in animals with sub-compensated diabetes mellitus (control 2) in all possible comparisons between the available groups (*p* < 0.01). The values of free and erosive surfaces of bone trabeculae and the osteoclast reaction in animals with sub-compensated diabetes mellitus were significantly higher (*p* < 0.01) compared with the same indicator in the jaw defect area in healthy animals (Table 1). Thus, the results of low activity of osteogenesis processes are demonstrated during the reconstruction of a bone cavity of critical size against the background of diabetes mellitus. Characteristic signs of weak regeneration 30 days after surgery include low total bone area and trabecular thickness in combination with a wide eroded surface and multiple lacunae, as well as low osteoblastic reaction in combination with extensive proliferation of connective tissue and isolated signs of the formation of foci of bone beams. Late formation of bone trabeculae is combined with a large free surface area and low cellular load, which contributes to a slower rate of bone tissue regeneration in the lesion compared to healthy animals (Figure 2B, Table 1).

### 3.2. Regeneration of the Bone Cavity Walls Using a Collagen Sponge

Implantation of a collagen sponge into the defect area shows that the processes of osteogenesis in animals of this group (control group 3) are more active than in animals with sub-compensated diabetes mellitus (control group 2). In animals with implantation of a collagen sponge after the same observation period (4 weeks), a high-volume density of bone tissue was revealed—56.4[55.3;57.7]% (*p* < 0.01). The following results were also observed: a greater thickness of bone trabeculae—0.13[0.12;0.15] mm; higher osteoblastic activity with less erosive surface, i.e., lower osteoclast activity—11.2[10.0;12.1]% (*p* < 0.01); and higher volume density of the osteoid—30.2[28.9;31.8]% (*p* < 0.01) (Table 1). Among the dense and well-developed connective tissue, signs of incomplete resorption of the foreign body are characteristic (Figure 3A). However, 4 weeks after surgery in control group 3, an active cellular inflammatory response remained around the collagen implant (Figure 3B).

### 3.3. Regeneration of the Bone Cavity in Healthy Animals when Filling with CH–SA–HA

Panoramic microscopy of the bone cavity walls of a group of healthy animals 4 weeks after surgical intervention with implantation of “CH–SA–HA” was characterized by a noticeable predominance of the bone component in the formed defect area of the lower jaw, and a larger proportion of the bone beams’ surfaces was free from cells (Figure 4A). On polarization microscopy, the bone trabeculae had a characteristic ordered and compact fibrous structure with a typical laminar yellow-green osteoid glow (Figure 5A).

The parameters of BV in the peripheral zone of control group 4 significantly exceeded those of control group 4 in the bone defect area with implanted bioconstruction (*p* < 0.01) (Table 1). In the central zone of the bone defect, signs of activation of the osteoblast reaction were recorded, as were the formation of mature connective tissue and the structure of the cortical plate (Figure 4A).

The peripheral zone of the bone defect is filled with osteoblasts, and the volume of maturing connective tissue increases in the bone trabecular structures (Figure 4B). Bone trabeculae are embedded in immature connective tissue. Osteoclasts are detected in the forming inter-trabecular spaces, and signs of active ring coating of bone trabeculae by osteoblasts are characteristic (Figure 4C). The number of bone trabeculae with a predominantly free surface and intertrabecular spaces increases (Figure 4D). The peripheral zone of the bone defect indicates the growth of maturing connective tissue (Figure 4E).

### 3.4. Bone Defect Regeneration in Animals with Sub-Compensated Diabetes Mellitus under CH–SA–HA Implantation

During microscopic examination of a bone defect in animals with sub-compensated diabetes mellitus upon implantation of CH–SA–HA, signs of inflammatory infiltration in the walls of the defect and in the peripheral zone are usually not recorded. The histoarchitecture of the peripheral bone tissue zone in animals of this group was characterized by an ordered arrangement of bone beams with a predominance of free surfaces (Figure 5A). Also, in the peripheral zone of the jaws, a pronounced reaction from the bone marrow was found, characterized by a predominance of intermediate and mature forms of cells of the granulocytic series (Figure 5B). In the area of the postoperative defect in the studied animal group, the bone component, represented by the structures of cancellous bone, predominated (Figure 5C).

Bone trabeculae in the postoperative defect area did not have a clear spatial orientation. Most of the surfaces of the bone beams were free of cells. However, in some fields of view, flattened and cubic osteoblasts as well as a few osteoclasts with the formation of lacunae with varying severity were found on the surfaces. In the jaw defect area, areas of bone marrow formation with a predominance of intermediate and mature forms of cells of the granulocytic series were also identified.

The distribution nature of the studied BV variable of bone tissue in the area of the postoperative defect and in the peripheral zone corresponded to normal, which was confirmed by the Shapiro–Wilk tests (*p* > 0.05), as well as by visual graphic evaluation (Figure 6A).

Based on one-way analysis of variance, a significant influence of the group membership factor (*p* < 0.01) on the variable value of BV was noted. At the 4-week period, BV values in control 4 (CH–SA–HA) in the area of the postoperative defect were significantly higher (*p* < 0.01) compared to control 1 (under the blood clot) (Figure 6A).

The nature of the distribution of the bone trabecula thickness (BTT) in all the obtained samples corresponded to the Gaussian distribution, which was confirmed by Shapiro–Wilk tests (*p* > 0.05), as well as visual graphical assessment (Figure 6B). Based on one-way analysis of variance, a significant effect of the group membership factor after 4 weeks of the experiment (*p* < 0.01) on the BTT variable was noted (Table 1). Pairwise comparisons of the BTT parameters in the area of the bone defect between groups of control 1 and control 4 animals using the Welch *t*-test did not reveal significant differences (*p* > 0.1). There were significant differences (*p* < 0.01) between the BTT values of the bone defect zone and the periphery zone in control group 1 at 4 weeks.

The type of distribution of variable intertrabecular spaces (ITS) did not differ from normal, which was confirmed by Shapiro–Wilk tests (*p* > 0.05), as well as by visual-graphic assessment (Figure 6C). In the one-way statistical analysis (ANOVA) with three subgroups, controls 1 and 4 (peripheral and central zones of the bone defect) showed a significant influence (*p* < 0.01) of the group membership factor in each experimental group. Paired comparisons between control groups 1 and 4 revealed significantly lower ITS values in the implanted biopolymer group (*p* < 0.01). The ITS indicator in control 4 did not change when comparing the central and peripheral zones of the defect (Table 1). Analysis of correlations between BTT and ITS showed that in control group 1 and control 4 there are also no significant connections (*p* > 0.1). The obtained results indirectly indicate the asymmetry of the distribution of these parameters relative to each other under conditions of reparative regeneration of bone tissue. Similar results were obtained when comparing other variables (*p* > 0.05). The histomorphometric parameters of BV, BTT and ITS, which characterize regenerative processes both in the area of the bone defect and in the peripheral zone, demonstrate variable values with opposite vectors. However, the three-dimensional picture of the distribution of these values over the 4-week period of the experiment is disordered, and no significant differences in the distribution were found.

At 4 weeks, the OBS values in control group 4 remain higher compared to control group 1 and are 5.4 times higher than the osteoblast surface at the periphery of the bone defect (*p* < 0.01) (Table 1). Thus, the values of the continuous variable OBS in the group with implanted “CH–SA–HA” biopolymer were significantly higher compared to both control 1 and in comparison with the peripheral zone of control group 4 animals (*p* < 0.01) (Figure 6D).

The distribution of the histomorphometric criterion for assessing bone tissue, the “volume of eroded surface” (ES), did not differ from the normal one in most of the formed samples, which is shown in the graph (Figure 6E), as well as in the Shapiro–Wilk test (*p* > 0.05).

Pairwise comparisons using the Welch *t*-test 1 month after the formation of the bone cavity in control group 1 showed a significant (*p* < 0.01) effect of the group factor on the variable value of ES. The area of bone beams of osteoclasts significantly exceeded that in control group 4 (*p* < 0.01). The volume of lacunar surfaces in the peripheral zone was significantly less than that in the area of the bone defect (*p* < 0.01).

The stereometric parameter of bone tissue covered with osteoid was assessed by polarizing microscopy and recorded on the basis of a homogeneous yellow-green glow on the surface of the bone beams (Figure 5A). The distribution of the presented variable in all samples did not differ from normal, which is clearly demonstrated by the graphic image (Figure 6F), as well as by the Shapiro–Wilk test (*p* > 0.05).

In pairwise comparisons between healthy control groups 1 and 4, significant differences (*p* < 0.01) were noted at 4 weeks of follow-up, with higher values in the implanted CH–SA–HA biopolymer group. When comparing the volume of the osteoid surface of the peripheral region with the area of the bone defect with the implanted biopolymer, significantly lower values were noted in the indicated period of the experiment (Table 1).

The stereometric criterion of “free surface” of bone trabeculae not occupied by osteoblasts and osteoclasts (in percent) was calculated by simply subtracting the indicators of eroded and osteoblastic surfaces from the total bone surface index. The distribution of the variable was consistent with a Gaussian distribution in all samples, as evidenced by the free-surface density plot (Figure 6G) and the Shapiro–Wilk test (*p* > 0.05). The values of the “free surface” variable with elements of descriptive statistics are shown in Table 1.

Correlation analysis between histomorphometric variables characterizing the surfaces of bone trabeculae did not show significant relationships (*p* > 0.1).

Thus, filling a bone cavity of critical size in the maxillofacial region with the help of the CH–SA–HA construct in healthy rats leads to the early formation of spongy and compact bone neoformation signs. The activity of bone formation significantly exceeds the results in control group 1 in healthy animals.

The results of the study show that the use of the CH–SA–HA construct in diabetes mellitus creates a high efficiency of bone regeneration and significantly compensates for the level of osteogenesis. As can be seen, in the graphic images below (Figure 7), the nature of the distribution of the obtained variables is similar to the normal distribution. One-way analysis of variance, conducted between variables with equal variances, established the significance of the group membership influence in relation to each histomorphometric variable (*p* < 0.01).

The results of a comparative morphometric analysis of the bone cavity in control 2 and group 5 of animals showed a great advantage when implanting the proposed CH–SA–HA design. All morphometric criteria used in the present study indicate more active osteogenesis 4 weeks after surgery using the CH–SA–HA construct (Table 1). In animals with sub-compensated diabetes mellitus, upon implantation of the CH–SA–HA construct, in contrast to control animals with diabetes mellitus, a higher percentage of newly formed bone tissue was noted, as was the presence of greater thickness of bone trabeculae with smaller intertrabecular spaces and greater osteoblastic activity with a smaller erosive surface of the trabeculae. When compared with a group of healthy animals (controls 1 and 4), it can be stated that the process of osteogenesis compensation when using the CH–SA–HA construct in sub-compensated diabetes mellitus is at a fairly high level.

It is important to note that with the development of an active inflammatory process in the model of diabetes mellitus in rats, the peripheral zone of the bone defect (at a distance of 3 mm from the bone cavity), despite its isolation from the bone cavity and completely different numerical characteristics of bone structures, actively reacts in the form of a total decrease in the level of osteogenesis (Table 1, columns 7 and 8).

## 4. Conclusions

Significance of the obtained results from the perspective of revealing the mechanisms of osteogenesis: The results of the study showed that during an attempt to reconstruct of a critical-sized bone defect in the maxillofacial region, the dynamics of bone formation disorders under conditions of hyperglycemia indicate the fundamental possibility of osteogenesis activation when using structures based on modified polysaccharides. It seems appropriate to address the issue of creating specialized technologies and technical means in relation to obtaining osteogenic matrices for direct transplantation into bone tissue defects. Attention should be paid to the processes of managing the start of early osteogenesis through primary early critical-local angiogenesis in the maxillofacial region. The choice of the optimal carrier for the culture of osteogenic cells is one of the key stages in the creation of a tissue-engineered equivalent of bone tissue. The absence of a well-perfused module in the affected area in the early stages after extensive bone tissue trauma and the lack of ability of synthetic materials that do not contain growth factors and osteogenic cells to cover a large area of a critical-sized bone defect [44] are some of the main problems of long-term restoration of the integrity of spongy and compact bones. The state of hyperglycemia already 4–5 weeks after the modeling of sub-compensated diabetes mellitus causes gross changes in the formation of spongy and compact bones. This is seen to a greater extent in the surgical intervention area and to a lesser extent on the periphery of the inflammation process.

Hyperglycemia and bone formation processes: In hyperglycemia, low bone regeneration and periodontal loss are caused by the formation of high concentrations of hydrogen peroxide (H_2_O_2_) in cell organelles. Overproduction of peroxides leads to differentiation and proliferation of osteoclasts [45,46,47]. Increasing the concentration of H_2_O_2_ in osteoclasts accelerates the differentiation and survival of these cells. Osteoclast activity weakens the structure of mineralized bone beams [48,49], disrupts ossification [50] and activates osteoblast apoptosis. This process leads to erosion of the bone beams [45,46].

Attention should be paid to the creation of conditions for early endothelialization of the matrix, i.e., the translation of the vascular endothelium from the walls of the bone cavity into the polymer structure. The formation of micro-vessels is the basis for the physiologically active process of bone formation [45]. Early local angiogenesis means an early start of osteogenesis mechanisms. Tight contact of the initially cell-free copolymer matrix with spongy and compact plates of the bone cavity with a certain filling of polymers with angiogenesis products triggers the process of matrix endothelization. Thus, early endothelialization of the artificial matrix is the primary task and will lead to the oriented sprouting of precursors or specialized cells, as well as to signaling molecules of intercellular interaction, and, as a result, an active process of early osteogenesis starts [46]. Existing studies have shown that despite the successful solution to the issue of early bone formation, the first stage of the process of reconstruction of the vasculature remains insufficiently understood from the point of view of the contact interaction mechanism between the vascular endothelium and artificial or natural polymers. It is known that the stimulation of proliferation and translation of the vascular endothelium occurs outside the vascular wall in the tissue compartment [47,48]. If polymer matrices are used, during the degradation of which nanoparticles are formed, this enhances the efficiency of vector molecules’ delivery to cells and leads to overexpression of angiogenesis molecules, enhancing endothelial recruitment and the new formation of blood micro-vessels [49]. Thus, angiogenesis occurs earlier than new bone formation [50]. Using this postulate, it should be clarified that the formation of micro-vessels in the matrix body is also an earlier stage of reconstruction, followed by osteogenesis [47,51,52]. The initial start of endothelialization ends with the rapid filling of the cell-free matrix with the vascular network. The osteoblastic and osteoclastic microenvironment of the vascular implant stimulates the rapid formation of spongy and compact bone from the periphery to the center of the implant. It is very important that the artificially created vascular network does not regress after complete disposal of the implant. Uncovering the mechanisms of the initial stages of angiogenesis will solve the problem of critical early osteogenesis.

The role of collagen in the reconstruction of a bone defect in combination with polysaccharides (chitosan, alginate): As shown by the present studies, collagen is able to improve the morphological and histological characteristics of the bone in the maxillofacial region in animals with induced diabetes mellitus. However, it is known that the use of pure collagen is not designed to directly activate angiogenesis. The inclusion of chitosan in the collagen complex triggers the formation of the vascular network [53]. It can be assumed that the collagen-chitosan copolymer is a promising material for the reconstruction of large bone defects.

The addition of a standard apatite component to the chitosan base, which is used for plasty in the maxillofacial region in patients [54,55]: The use of a bone apatite matrix intermediate, octacalcium phosphate (Ca_8_H_2_(PO_4_)_6_ 5H_2_O), is an attempt to obtain higher bone regeneration activity and create an alternative to autologous bone grafting [56]. It is fair to clarify that the same regularity is also revealed when the alginate-hydroxyapatite framework is combined with poly-lactic acid [57,58], ethyl-cellulose and poly(ε-caprolactone) [59]. In general, the introduction of additional natural polysaccharide polymers into the structure leads not only to the obtainment of excellent mechanical strength, but also to the active implementation of cellular functions such as adhesion, proliferation and differentiation, which is very important. For example, a composite scaffold consisting of a combination of chitosan, alginate and hydroxyapatite and cellulose nanocrystals possesses such properties, which is especially valuable for bone tissue engineering.

Attention should be paid to the use of sulfated polysaccharides, such as heparin, in the study of the preparation of the functional copolymer. Three sulfate groups in the heparin molecule allow for large, rigid, linear polymers to be transformed into nanosized globules, which predetermines their high mobility in living animal tissues [31]. It is noteworthy that active electrostatic interaction between the negatively charged sulfate groups of heparin and the positively charged amino acid residues of the growth factor leads to an increase in binding affinity [60]. An additional positive feature of the use of heparin is that its low molecular weight allows for a large amount of polysaccharide to be loaded into a large carrier molecule. Such a polymer network becomes resistant to enzymatic hydrolysis. It is assumed that the high affinity of heparin for growth factors provides a high protein load, for example, for the “adgelon” serum growth factor used in this work. The higher the growth factor load, the greater the angiogenesis effect. It is very important that such a growth factor incorporation technology with a good final functional result can be applied by conjugating two or more polysaccharides. The introduction of heparin into high-molecular scaffolds based on alginate and chitosan has good mechanical properties, non-toxicity, low immunogenicity and the ability to regulate the rate of degradation [61].

The role of modified chitosan in bone cavity reconstruction: Studies have shown that modified chitosan in a case without diabetes can significantly improve the histomorphological characteristics of the bone cavity walls in the maxillofacial region. It is known that the use of a chitosan membrane as a single polymer leads to the formation of new bone and cementum in single-wall intraosseous defects in large experimental animals [62]. The combined use of biopolymers in one structure is designed for the mutual penetration of individual polymers with the hydrogel network’s formation, which creates reinforcement of the structure and is considered one of the best technologies for obtaining a composite with high mechanical properties [63].

Similar combined meshes as those used in our work are known for the purpose of bone tissue bioengineering, with the inclusion of alginate and chitosan [64] or hyaluronic acid [65,66] in the design. The low penetrating ability of chitosan as an independent molecule through the inflammatory tissue compartment can be significantly increased by creating a complex polyelectrolyte structure, which leads to the densification of the polymer to nanosizes. It has been established that upon passing from pH 1.55 to pH 3.5, the size of nanoparticles decreases from 200–220 nm to 48–52 nm due to a pH-dependent change in the conformational state of the molecule. Most likely, in the environment of the stratified epithelium of the oral cavity, where the pH is close to 5.5, the nanometer scale of molecular sizes is a fundamental factor for the chitosan use. The design, which contains high-purity and deacetylated chitosan, sodium chondroitin sulfate, sodium hyaluronate, heparin sulfate, sodium alginate and amorphous hydroxyapatite in certain weight ratios, is designed for early stimulation of bone formation. It is known that a nanosized surface, in comparison with a micro-sized one, is a stimulus for the proliferation of endothelial cells and osteogenic cells [38]. It is important to note that a high degree of deacetylation of the chitosan matrix plays an important role both in terms of structure degradation and in the degree of its endothelization [67]. An important intermediate goal when introducing a polyelectrolyte construct into a bone cavity of critical size is to increase the efficiency of the polysaccharide implant structure’s contact with the bone walls and the microvascular bed of the bone in a shorter time after the intervention. This goal is achieved due to the fact that a nanogel mass of chitosan-alginate-hydroxyapatite (CH–SA–HA) is introduced into large cavities. These studies are an example of experimental modeling of the real situation of protection of small molecules in aggressive environments, for example, the oral cavity, and they are directly related to solving the problem of targeted medicinal substances’ transportation fixed on chitosan.

The use of a combination of chitosan with an alginate–hydroxyapatite framework [68,69,70,71] stabilizes the gel matrix due to the formation of polyelectrolyte complexes. The complexes are formed as a result of a weak interaction of oppositely charged chemical groups of COO- alginate and NH_3_^+^ of chitosan. In addition, NH_3_^+^ groups can interact with PO_4_^3−^ hydroxyapatite groups, linking the framework together and forming a more compact structure. At the same time, the porosity and degradation rate decrease, and mechanical stability increases. If we are talking about the destruction of periodontal tissues in the experiment, then the filling of a bone defect with a chitosan complex with a polyanionic structure [72] leads to inhibition of the apical migration of the epithelium and active formation of new bone and cementum. Thus, there is a need to use such constructs to close critical bone defects in diabetes mellitus, given the well-known scientific findings that reveal the mechanisms of activation of osteogenesis and angiogenesis. Biodegradable polysaccharide biopolymers as the main substrates for self-regulation, as well as for the purposes of overexpression of osteogenic and angiogenic growth factors and the activation of mesenchymal stem cells, osteoblast precursors and vascular endothelial cells, can be considered as promising constructs in the treatment of a specific inflammatory process in the periodontium. Solving this problem is a complex project, since the systemic impairment of bone formation in diabetes mellitus affects extracellular, transcellular and intracellular mechanisms associated with osteoinduction and osteoconduction. The complexity of the regulation of the cytokine storm and the development of osmotic and oxidative stress in cells with the accumulation of peroxides requires the use of complex building structures for contact with the maternal bone that can interfere with the processes of osteoclast activation, bone resorption, osteomalacia and prolonged inflammation in the periodontal zone. In this regard, polysaccharide biopolymers, such as chitosan or alginate, are very promising for bone bioengineering. They can independently or in the presence of exogenous growth factors increase the quality and volume of mineralization, encourage the formation of bone beams with a large mass of osteoblasts and osteoids and increase the total number of osteocytes in the newly formed bone. It has been established that in in in vitro and in vivo systems with the presence of chitosan and alginate in combination with nano-hydroxyapatite, the growth of angiogenesis factors (VEGF, CD31) in osteoblast precursors and mature osteoblasts is detected early. The active proliferation of osteoblast precursors, the formation of a large mass of osteocytes filling a critical-sized bone defect and microvascular endotheliocytes in Haversian canals confirm the wide functional nature of modified chitosan implants in relieving the inflammatory process in induced diabetes mellitus.

Thus, 4 weeks after the creation of a critical-sized bone defect, the proposed model of induced type I diabetes mellitus in rats leads to distinct morphological disorders of osteogenesis both in the surgical intervention area and in the lower jaw peripheral zone. Such disorders are characterized by activation of the osteoclastic reaction and erosion of the bone walls, inhibition of the osteoblastic reaction during the bone trabeculae formation and an increase in free bone surfaces depleted in the cell mass of various functional directions. The presence of type I diabetes mellitus in rats contributes to a slowdown in the rate of bone tissue regeneration in the lesion compared to healthy animals, as in cases with the implanted CH–SA–HA biopolymer. Implantation of a polyelectrolyte polysaccharide complex based on “CH–SA–HA” modified chitosan in the lower jaw region of a formed critical-sized bone defect in healthy rats and rats with induced type I diabetes mellitus promotes a significantly faster rate of bone tissue repair. The best results of the inflammatory process resolving and restoring the bone were obtained in the absence of diabetes mellitus and cavity reconstruction using the CH–SA–HA complex as early as 4 weeks after surgical intervention. The use of a lyophilized collagen sponge based on type I collagen for the purpose of bone cavity reconstruction reliably indicates the activation of bone formation, despite the lower rates compared to the results of using the CH–SA–HA complex, and can be used in combination with modified chitosan. The proposed CH–SA–HA design for diabetes mellitus demonstrates more active osteogenesis 4 weeks after surgery. This result consists of a higher percentage of newly formed bone tissue appearing, the presence of greater thickness of bone trabeculae with smaller intertrabecular spaces and high osteoblastic activity with a smaller erosive surface of the trabeculae. The process of osteogenesis compensation when using the CH–SA–HA construct in sub-compensated diabetes mellitus is at a fairly high level. Earlier studies of the polysaccharide copolymer in clinical settings confirm the effectiveness of the design in the treatment of periodontitis in patients suffering from diabetes mellitus, especially with mild to moderate tissue damage. The promising use of the design in patients to close bone defects of a critical size after cystectomy or removal of impacted teeth creates the opportunity to expand the arsenal of treatment technologies in dentistry. A limitation to the use of CH–SA–HA in clinical settings in patients with type I diabetes mellitus is a significant decrease in the effectiveness of use as the severity of the disease increases, especially with severe periodontal damage.

The possibility of using the CH–SA–HA polysaccharide construct in a clinical setting in patients aged 18–35 years with type I diabetes mellitus and a diagnosis of chronic widespread periodontitis with a duration of no more than 3 years is cited in earlier studies [41,73]. It should be noted that the new product CH–SA–HA has previously passed official medical tests in GCP format for Russia in order to register the product in the register of medical devices of the Ministry of Health in the field of supervision. Preclinical sanitary-chemical and toxicological tests were carried out in an accredited testing laboratory center of the Research Institute of Physical and Chemical Medicine of the Ministry Health of the Russian Federation (protocol and conclusion No. 1612.007, dated 13 August 2007, and No. 1488.007, dated 19 July 2007). The regulatory document entitled “Wound covering based on the chitosan-alginate complex “CAH-BOL”, sterile”, has been approved. Technical specifications 9393-006-78455243-2008 were approved by the Federal State Institution “Institute of Surgery named after A.V. Vishnevsky of Russian Medical Technologies”. The production of the CH–SA–HA design was carried out on the basis of the Collachit Limited Liability Company, Zheleznogorsk, Krasnoyarsk region, Russia. Clinical trials of the design were carried out after the decision of the Ethics Committee of the Krasnoyarsk State Medical University (protocol No. 4, 30 November 2007) and the conclusion of a contract between the Krasnoyarsk Medical University and the municipal city clinical hospital No. 6 (Krasnoyarsk), 15 November 2007. The CH–SA–HA construct was inserted into periodontal pockets after curette treatment and polishing of the surface of the tooth roots. The CH–SA–HA polysaccharide implant was used in patients with compensated type I diabetes mellitus; a control was carried out, and positive results were obtained in the treatment of generalized periodontitis of mild and moderate severity. The conclusion of these studies is as follows: The prevalence of periodontal diseases in the examined patients with type I diabetes mellitus is 100%. The structure of inflammatory periodontal diseases is dominated by chronic generalized periodontitis of mild (group I) and moderate severity (group II) (88%). The use of the CH–SA–HA design in the treatment of mild generalized periodontitis provides a reduction in gingival bleeding (PBI, Saxer and Muhlemann index) by 4.5 times, in the inflammation index (papillary-marginal-alveolar index (PMA)—by 2.4 times and in the periodontal index (PI)—by 2.3 times compared to the control results with moderate severity of the disease, respectively—2.4 times, 1.4 times and 1.6 times. The anti-inflammatory effect in the first group of patients was 56%; in the second group—37%.

The use of the CH–SA construct (hydroxyapatite is not included in the copolymer molecule) in a clinical setting was performed in 30 patients aged 30–65 years after removal of a radicular cyst (15 people were operated on with implantation of a polysaccharide construct, and 15 people were used as controls, in whom the bone cavities were filled with a blood clot). All patients had bone defects of critical size after removal of the cyst. In addition, the polysaccharide implant was used in 13 patients aged 20–65 years after the removal of impacted teeth [74,75,76,77]. Studies have shown the high effectiveness of the CH–SA design in closing bone defects of critical size (Figure 8a–c).

For example, removal of a radicular cyst in the area of the 12th, 11th, 21st and 22nd tooth in a patient aged 46 years in the department of maxillofacial surgery of the regional clinical hospital of the Krasnoyarsk region was accompanied by filling the bone cavity using a CH–SA construct. Six months after implantation, the absence of destructive bone changes in the area of the defect of the right upper jaw was recorded. Densitometric examination determines the pronounced optical density of the bone regenerate when compared with a section of intact bone tissue. Thus, with a bone layer thickness of 10 mm (simulation of an intraoral radiograph), the density of the regenerate on the Hounsfield scale (HU) in the area of the former defect was 801 units, and it was 881 units in the area of bone tissue outside the border of the surgical field (Figure 9a,b).

During a densitometric study of a 5 mm thick bone tissue reformat, a fairly pronounced optical density of the regenerated defect was determined in comparison with the optical density of bone tissue outside the surgical field: in the area of the defect, the peak was 969 units on the HU scale, and in the symmetrical area of the opposite side—1106 units (Figure 9c,d).

Densitometric examination of a 1 mm thick section of bone tissue revealed a pronounced optical density of the regenerated bone defect compared to bone tissue outside the surgical field: in the area of the operation—550 units on the HU scale; outside the surgical field—650 units (Figure 9e,f).

Thus, the results of experimental studies on the use of modified chitosan with a high degree of deacetylation and high molecular weight as part of a copolymer with alginate and hydroxyapatite serve as the basis for the reconstruction of bone defects of a critical size in the maxillofacial area. Such use requires the selection of physical and chemical synthesis conditions, a solution to the problems of controlling angiogenesis processes and osteogenesis and the use of certified products in experimental and clinical practice.

## Figures and Tables

**Figure 1 polymers-15-04337-f001:**
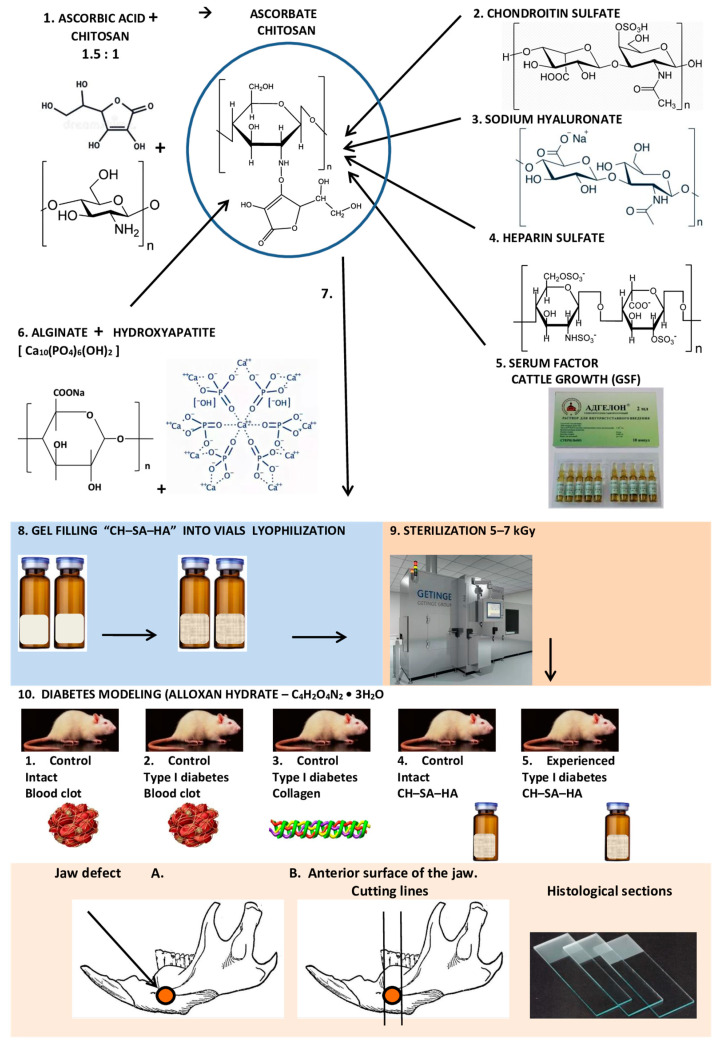
Study design.

**Figure 2 polymers-15-04337-f002:**
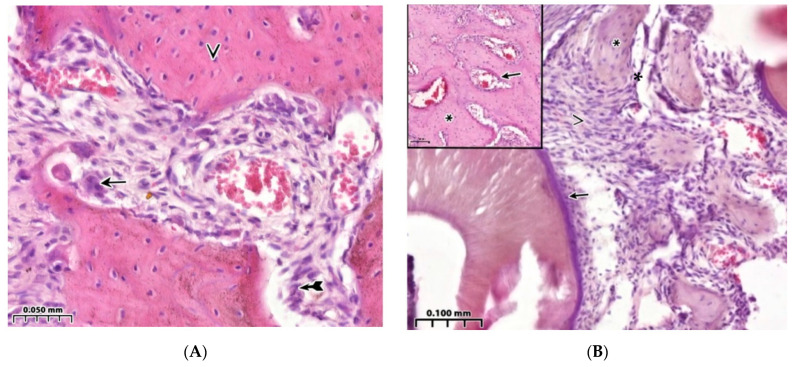
Control healthy animals (**A**) and animals with a model of diabetes mellitus (**B**). Healing of a bone defect under a blood clot. Hematoxylin-eosin staining. (**A**)—Cellular reaction in the bone regeneration zone. On the surface of the bone trabecula, there is an active osteoclastic reaction (↑) and a moderate number of osteoblasts (➼), as well as “isolated” osteocytes in the bone tissue (v). (**B**)—Wall of the bone cavity. Bone trabeculae have a predominantly free surface (∗) and are located among the extensive surface of connective tissue (^), and periodontal ligament is shown (↑). In the upper left corner, the structure of healthy bone is shown: formed bone tissue with compactly located bone trabeculae (*), narrow inter-trabecular spaces with bone marrow structures (↑).

**Figure 3 polymers-15-04337-f003:**
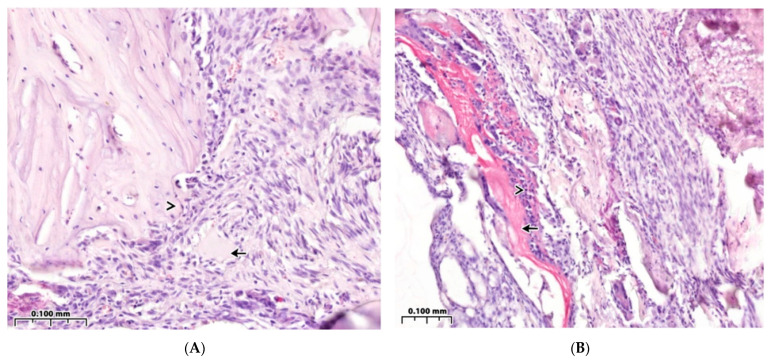
Control animals with a model of diabetes mellitus. Implantation of a collagen sponge into the bone cavity. Hematoxylin-eosin staining. (**A**)—Among the dense connective tissue, amphophilic structures of a foreign body (↑) and bone trabeculae with a free surface (^) are visible. (**B**)—Remnants of implanted collagen are visible (↑), as is cellular inflammatory infiltration around the implant (^).

**Figure 4 polymers-15-04337-f004:**
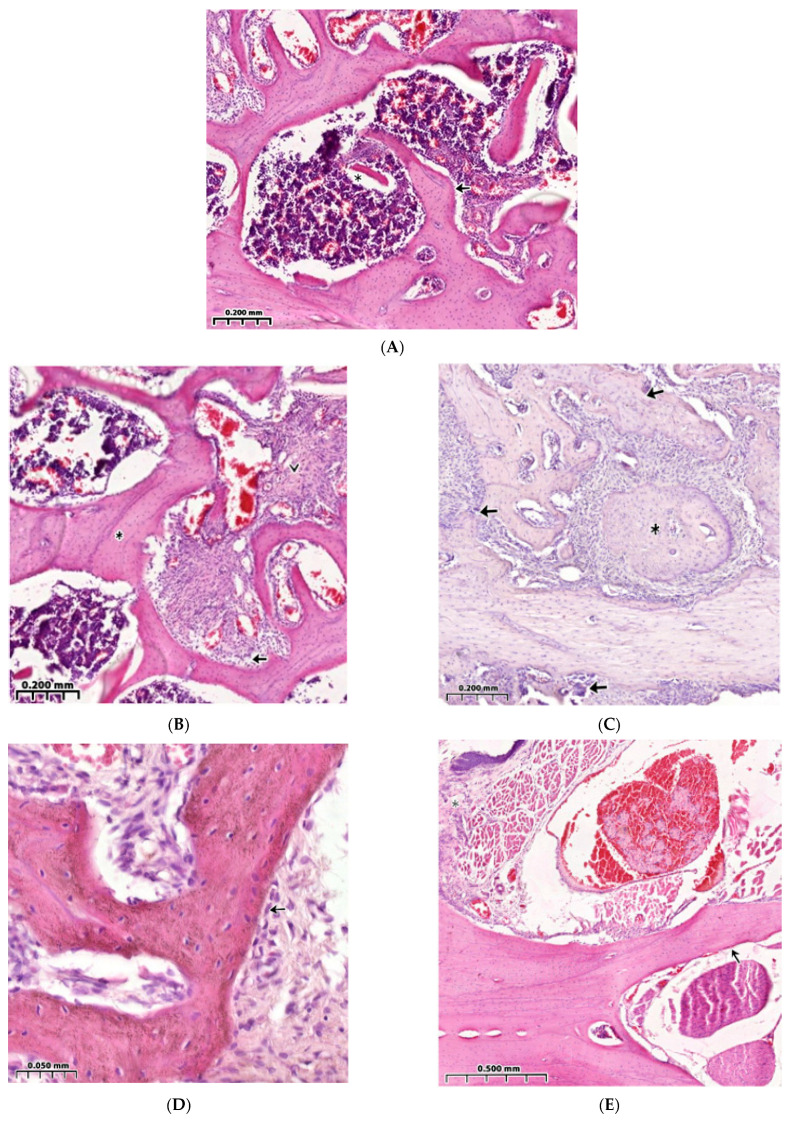
Healthy control animals. Implantation into the bone cavity of chitosan ascorbate-sodium alginate-hydroxyapatite (CH–SA–HA). Observation for 30 days. Hematoxylin-eosin staining. (**A**)—Significant predominance of regenerating bone tissue in the area of the formed defect, bone beams with a predominantly free surface (↑), and reactive bone marrow structures are visible in the inter-trabecular spaces between the beams (*). (**B**)—In the peripheral zone of the bone defect, osteoblasts are visible (↑), and maturing connective tissue (^) of bone trabecular structures (*) is recorded. (**C**)—Cellular reaction in the bone regeneration zone. A bone trabecula (*) is identified among the immature connective tissue; an osteoclast is visible in the resulting gap (↑). (**D**)—Active ring coating of bone trabeculae by osteoblasts (↑). (**E**)—Bone trabeculae with predominantly free surfaces (↑) and inter-trabecular spaces (^) and an adjacent peripheral area with proliferation of maturing connective tissue (*) are visible.

**Figure 5 polymers-15-04337-f005:**
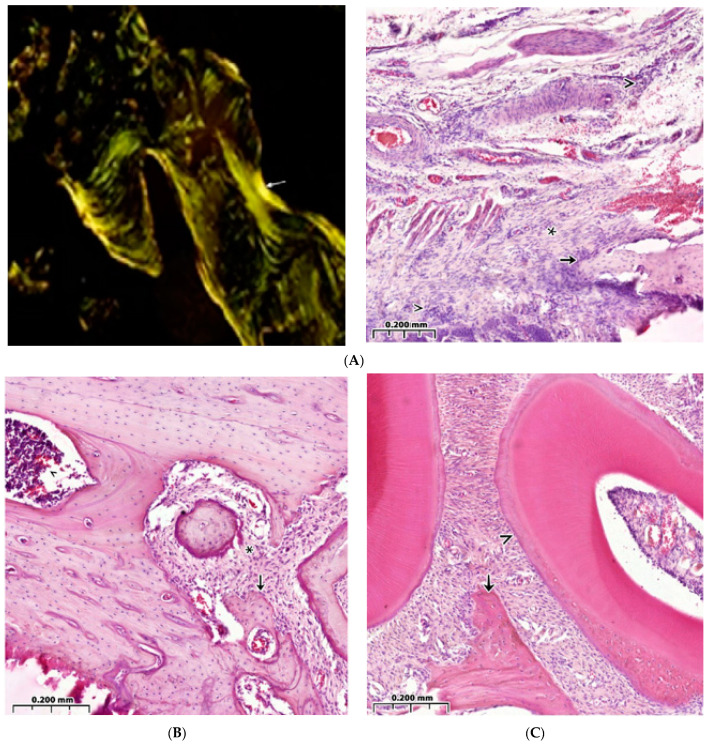
Animal models of diabetes mellitus. Implantation into the bone cavity of chitosan ascorbate-sodium alginate-hydroxyapatite (CH–SA–HA). Hematoxylin-eosin staining. (**A**)—Regenerative changes in the bone defect and peripheral zone. Bone beams with a free surface (↑). In the field of view, there is a section of the peripheral region (*) with moderate infiltration of histiocytes and fibroblasts, and in the inter-trabecular spaces, there are reactive bone marrow structures. (**B**)—In the peripheral border zone, there is a site of connective tissue maturation (*), as well as mature bone tissue with unpronounced inter-trabecular spaces (↑) and reactive bone marrow structures in the inter-trabecular spaces (^). (**C**)—Near the periodontal ligament and tooth root structures (^), there is mature bone tissue (↑), and there are no signs of the presence of polysaccharide structures (complete biodegradation).

**Figure 6 polymers-15-04337-f006:**
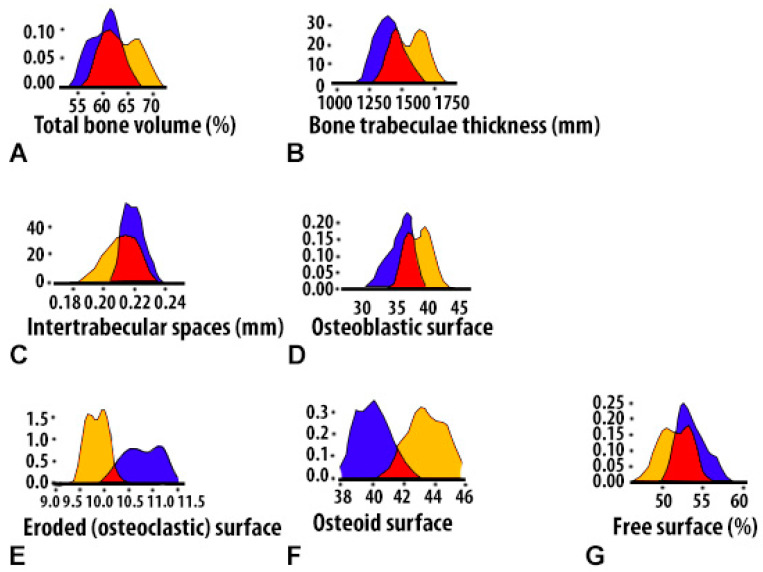
Histomorphometric criteria for density in the bone cavity of a critical size in healthy rats 4 weeks after implantation of the CH–SA–HA composition and blood clot: (**A**)—distributions of the studied variable bone density (BV); (**B**)—distributions of bone trabecula thickness (BTT) in the obtained samples; (**C**)—variable size distributions of inter-trabecular spaces (ITSs); (**D**)—density distribution of the OBS index in controls 1 and 2 in healthy animals after 4 weeks of observation; (**E**)—density distribution of the eroded (osteoclastic) surface in controls 1 and 2; (**F**)—density distribution of the osteoid surface in controls 1 and 2; (**G**)—free-surface density distribution in controls 1 and 2. The ordinate axis shows the density of bone formations. Under a blood clot—blue fragment of the graph, CH–SA–HA—yellow fragment of the graph.

**Figure 7 polymers-15-04337-f007:**
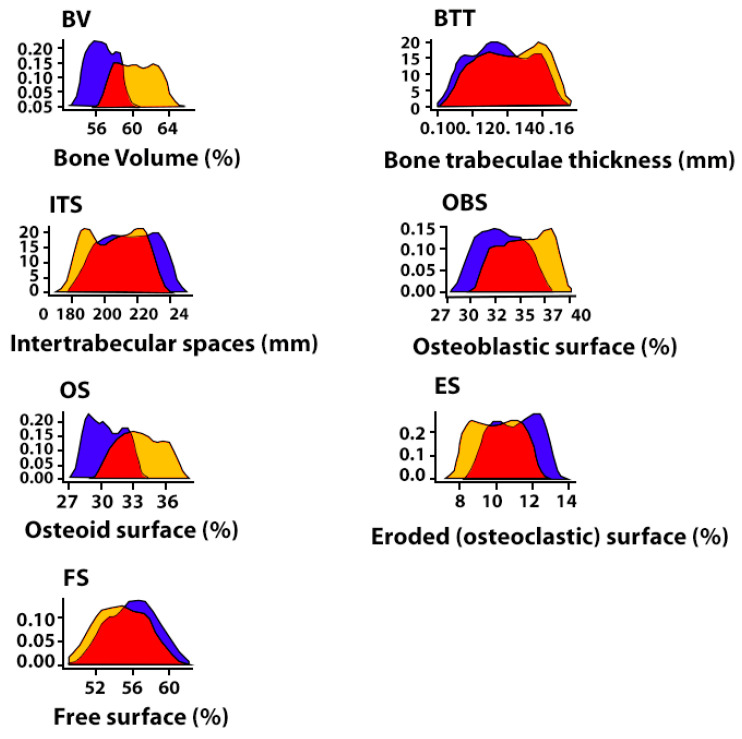
Distribution density of histomorphometric parameters in the area of postoperative bone tissue defect in rats with sub-compensated diabetes mellitus in control group 3 (collagen) and experimental group (CH–SA–GA) 4 weeks after modeling the disease. The ordinate axis shows the density of bone formations. Collagen—blue fragment of the graph, CH–SA–HA—yellow fragment of the graph.

**Figure 8 polymers-15-04337-f008:**
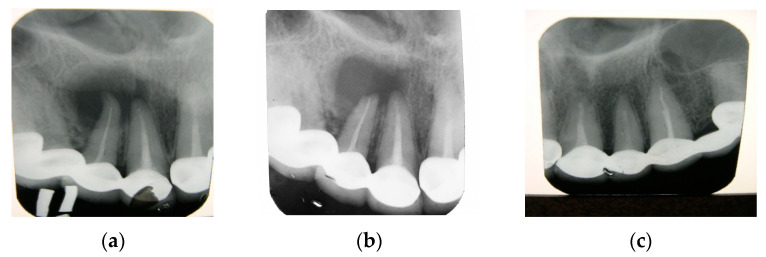
Radicular cyst of the patient’s left upper jaw (tooth 11.12): (**a**)—intraoral radiograph before surgery; (**b**)—intraoral radiograph 3 months after implantation of the CS–SA-structure, (**c**)—intraoral radiograph 6 months after implantation of the CS–SA-structure (complete restoration of the bone in the defect area, bone trabeculae are well traced).

**Figure 9 polymers-15-04337-f009:**
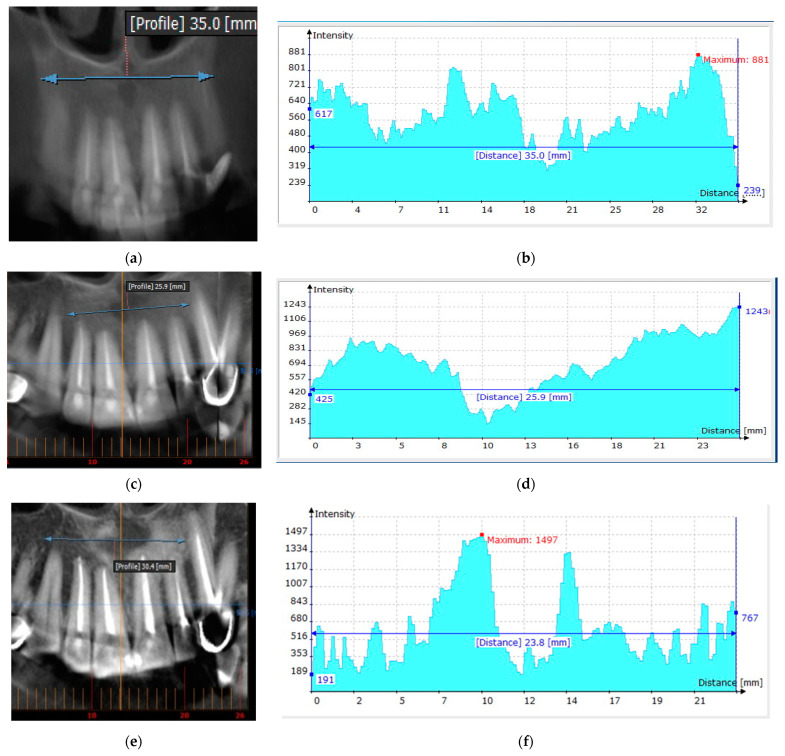
(**a**)—Sonogram of the frontal part of the upper jaw—reformatted cone beam computed tomography (PicassoTrio). Bone thickness 10 mm. (**b**)—Densitogram of the frontal part of the upper jaw. Bone tissue thickness 10 mm. (**c**)—Reformat of the frontal part of the upper jaw with cone beam computed tomography (PicassoTrio). Bone thickness 5 mm. (**d**)—Densitogram of the frontal part of the upper jaw. The thickness of the bone tissue is 5 mm. (**e**)—Cone beam computed tomography (PicassoTrio) reformat of the anterior maxilla, 1 mm thick. (**f**)—Reformat of the frontal part of the upper jaw with cone beam computed tomography (PicassoTrio), 1 mm thick.

**Table 1 polymers-15-04337-t001:** Histomorphometric criteria for bone reconstruction in sub-compensated diabetes mellitus under CH–SA–HA implantation (Me [25;75]).

Histomorpho Metric Criterion	4-Week-Old Mandibular Defect Area					Peripheral Zone of Bone Control 4 Healthy (CH–SA–HA)	Peripheral Zone of Bone (Diabetes Mellitus) Collagen + CH–SA–HA
	Control 1 Healthy (under the Blood Clot)	Control 2 (Diabetes Mellitus) (under the Blood Clot)	Control 3 (Diabetes Mellitus) (Collagen)	Control 4 Healthy (CH–SA–HA)	Experienced (Diabetes Mellitus) (CH–SA–HA)		
1	2	3	4	5	6	7	8
BV ^1^ (%)	60.9[58.0;62.0] **	50.4[49.5;51.3] +	56.4[55.3;57.7] ***	62.9[60.7;66.8]	60.3[58.5;62.1] †	66.5[64.3;68.1] ••	61.2[58.0;64.3] ++
BTT ^1^ (mm)	0.14[0.13;0.15] *	0.12[0.10;0.13] +	0.13[0.12;0.15] ***	0.16[0.15;0.17]	0.14[0.12;0.16] †	0.16[0.15;0.17] ••	0.14[0.13;0.16] ++
ITS ^1^ (mm)	0.22[0.21;0.22] **	0.20[0.19;0.21]	0.21[0.20;0.22] ***	0.21[0.21;0.22]	0.20[0.19;0.21] †	0.21[0.20;0.22]	0.18[0.17;0.20] ++
OBS ^1^ (%)	36.0[34.7;37.0] **	29.1[28.5;29.8] +	32.9[31.1;34.8] ***	38.9[37.1;40.1]	35.5[33.4;37.5] †	7.2[6.3;7.8] ••	4.5[3.9;5.3] ++
OS ^1^ (%)	40.0[39.1;40.7] **	27.1[26.0;28.2] +	30.2[28.9;31.8] ***	43.3[42.8;44.3]	33.5[32.0;35.3] †	10.3[9.6;11.0] ••	6.9[5.9;8.0] ++
ES ^1^ (%)	10.7[10.5;11.1] **	15.4[14.2;16.6] +	11.2[10.0;12.1] ***	9.9[9.7;10.0]	10.0[8.9;11.0] †	1.3[1.2;1.4] ••	1.3[1.1;1.4] ++
FS ^1^ (%)	53.1[52.5;54.7] **	62.0[60.3;63.7] +	56.1[54.0;57.8] ***	51.5[50.1;53.2]	54.6[52.7;56.9] †	91.4[90.9;92.6] ••	94.1[93.4;94.8] ++

* Differences are reliable with control 4 in the area of the bone defect (*p* < 0.05). ** Differences are reliable with control 4 in the area of the bone defect (*p* < 0.01). *** Differences are reliable with experienced group in the area of the bone defect (*p* < 0.01). •• Differences are reliable with the corresponding variable when comparing the histomorphometric criterion of the peripheral and central zones in animals of control group 4 (*p* < 0.01). ^1^ Differences are reliable in multiple-comparisons ANOVA (*p* < 0.001). † Differences are reliable when comparing the corresponding variable values in animals with diabetes mellitus in control group 2 (*p* < 0.01). + Differences are reliable with the corresponding variable in animals with diabetes mellitus in control group 2 when compared with control 4 and experienced (*p* < 0.01). ++ Differences are reliable with the corresponding variable in animals in the peripheral zone of the bone in control group 3 and the experimental group when compared with the experimental group in the zone of the bone defect (*p* < 0.01). BV—volumetric density of bone tissue; BTT—thickness of bone trabeculae (mm); ITS—intertrabecular spaces (mm); OBS—osteoblastic surface of bone trabeculae; OS—osteoid surface of bone trabeculae; ES—eroded (osteoclastic) surface of bone trabeculae; FS—free surface of bone trabeculae.

## Data Availability

Written informed consent was obtained from patients for the use of the CH–SA polysaccharide construct for the treatment of bone defects of critical size after removal of impacted teeth or cystectomy, as well as the use of the CH–SA–HA construct in patients with a generalized form of chronic periodontitis associated with type I diabetes mellitus.

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
