# Peer review of "Morphological Reconstruction of a Critical-Sized Bone Defect in the Maxillofacial Region Using Modified Chitosan in Rats with Sub-Compensated Type I Diabetes Mellitus"

_polymers, 2023, doi:10.3390/polym15214337_

Round 1

Reviewer 1 Report

Comments and Suggestions for Authors

The manuscript exhibits shortcomings in its writing quality, particularly noticeable in the disorganized presentation of figures. The logical flow appears chaotic and bewildering, deviating from the expected structure of an academic paper. Notably, the absence of radiology methods, such as microCT, to substantiate claims for bone regeneration – a gold standard in this context – raises questions about the author's approach. It is advisable to reference established academic papers to gain insights into the characteristics that define a truly rigorous scholarly article.

Comments on the Quality of English Language

Extensive editing of English language required

Author Response

Notes are included in the text of the article

Reviewer 2 Report

Comments and Suggestions for Authors

In this manuscript, authors have synthesized a material mainly using chitosan along with  other biocompatible substances chondroitin sulfate, sodium hyaluronate, heparin, sodium alginate, etc. This material was tasted on the rats to see its functionality towards bone regeneration. The subject of the manuscript is promising and it is suitable for publication in Polymers. I have following suggestions to improve the quality of the manuscript. 

1. In introduction it is advised the authors to include a discussion of their published article (Biomed Transl Sci. 2022; 2(1):1-8) with the relevance to this work.

2. A thorough English revision is also recommended. 

3. Resolutions of figures should be improved. 

4. A discussion related to the possibility of the implementation of this work in the human body should also be included. 

5. A brief conclusion of the study containing its importance and limitations should also be included in the manuscript. 

Comments on the Quality of English Language

There are some minor grammatical mistakes that should be corrected. 

Author Response

(The authors gave the same response as above.)

Reviewer 3 Report

Comments and Suggestions for Authors

Dear authors

I cannot accept this manuscript in this form.

1) You never mention what adgelon is and why you use it..

2) In the Materials and Methods specify when the samples are collected. I have doubts about regeneration after 4 weeks, particularly on a critical size lesion. A kinetics would be indicated.

3) HE staining is not enough to show regeneration; for example, use a Gomori trichrome staining and microCT which allows bone regeneration to be clearly seen. Indicate the presence of blood vessels. Place a bone structure as a control. Locate the section in relation to the lesion. Avoid putting one image per figure. Show clearly regeneration in the center and on the edges of the lesion. There are no annotations on the figures, put a scale bar which is better than the magnification. Why don't all the figures have the same color?

4) Figure 18 to review, what is the point of this figure? Why do a Safranine O staining, this staining is not indicated in the Materials and Methods.

5) Why are there grayed out paragraphs in the text?

Sincerely

Author Response

(The authors gave the same response as above.)

Round 2

Reviewer 3 Report

Comments and Suggestions for Authors

Dear authors

You have answered the various questions. I still have a few comments.

What does the gray highlighting mean in the text?

In Figure 2, the 3 photos should be separated and annotated A, B and C for better understanding. In Figures 2 and 6 to 15 the annotations are not clearly visible.

If the modifications are made, this manuscript may be published in Polymers.

Best regards

Author Response

The authors of the article thank the reviewer for their expertise and posing specific questions, and the opportunity to improve the quality of the research results.

Journal: Polymers
Manuscript ID: polymers-2584383
Title: Morphological reconstruction of a critical size bone defect in the
maxillofacial region using modified chitosan in rats with sub-compensated
type I diabetes mellitus
Authors

Igor N Bolshakov * , Nadezhda N. Patlataya , Vladimir A. Khorzhevskii , Anatoli A. Levenets , Nadezhda N Medvedeva , Mariya A Cherkashina , Matvey M. Nikolaenko , Ekaterina I Ryaboshapko , Anna E. Dmitrienko

Section

Polymer Composites and Nanocomposites

Special Issue

Chitosan-Based Polymers as Promising Materials for a Variety of Biomedical Applications

The authors of the article answered the questions posed by reviewer 3 and uploaded the answers along with the revised article.

Answer: 1. The selections you specified in the text are excluded (using computer keys: home - clear format). 2. In Figure 2, a boundary is drawn between two illustrations (diabetes mellitus model and healthy bone). Figures 2 and 6 to 15 have more visible symbols.

Best regards, Prof.IN Bolshakov 

04/10/2023
